# TransConv: Transformer Meets Contextual Convolution for Unsupervised Domain Adaptation

**DOI:** 10.3390/e26060469

**Published:** 2024-05-29

**Authors:** Junchi Liu, Xiang Zhang, Zhigang Luo

**Affiliations:** School of Computer Science, National University of Defense Technology, Changsha 410073, China; zhangxiang08@nudt.edu.cn (X.Z.); zgluo@nudt.edu.cn (Z.L.)

**Keywords:** transformer, convolution, unsupervised domain adaptation, contextual information

## Abstract

Unsupervised domain adaptation (UDA) aims to reapply the classifier to be ever-trained on a labeled source domain to a related unlabeled target domain. Recent progress in this line has evolved with the advance of network architectures from convolutional neural networks (CNNs) to transformers or both hybrids. However, this advance has to pay the cost of high computational overheads or complex training processes. In this paper, we propose an efficient alternative hybrid architecture by marrying transformer to contextual convolution (TransConv) to solve UDA tasks. Different from previous transformer based UDA architectures, TransConv has two special aspects: (1) reviving the multilayer perception (MLP) of transformer encoders with Gaussian channel attention fusion for robustness, and (2) mixing contextual features to highly efficient dynamic convolutions for cross-domain interaction. As a result, TransConv enables to calibrate interdomain feature semantics from the global features and the local ones. Experimental results on five benchmarks show that TransConv attains remarkable results with high efficiency as compared to the existing UDA methods.

## 1. Introduction

Deep neural networks have achieved impressive success in a wide range of computer vision applications [1], but their success usually demands massive quantities of labeled data for better representations. This often follows the assumption that training and testing sets are from the same data distribution. Nevertheless, this situation does not always work well in practice. One way out could be resorting to the unsupervised domain adaptation (UDA), which trains deep neural network models on rich labeled data from a related source domain. But this supervised learning suffers from the domain shift issue, resulting in poor generalization performance on other new target domains. To address this issue, considerable research efforts are devoted to such UDA tasks [2,3,4,5] by bridging the distribution discrepancy, minimizing distance metrics or adversarial learning, etc. In such arts, most existing approaches advance convolution neural networks (CNNs)-based frameworks to learn the domain-invariant feature representation. Such features are often from local receptive fields.

With the success of transformers in various visual tasks, recent UDA methods focus more on global features by using encoder–decoder frameworks, in contrast to local features learned by CNN frameworks. The most advanced domain adaptation methods extract global features of images by using transformer architecture as backbone network. Recent studies show that the models with transformers are obviously better than those with pure convolutional neural networks. For example, transferable vision transformers (TVT) [5] utilize the transferability adaptation module of vision transformers (ViT) [6] for domain adaptation. Cross-domain transformers (CDTrans) [4] use the robustness of cross-attention in transformers to propose a three-branch transformer model for UDA tasks. To take full advantage of both transformer and CNN architectures, a natural idea is combining both of them. However, CDTrans uses a two-stage training method, which takes a long time and is not conducive to the rapid migration of the model. The challenge of hybrid models is how to maintain the robustness of cross-attention with high efficiency.

On one hand, we introduce a Gaussian attended MLP module to further empower the robustness of the encoder of transformer by adjusting more attention to major channel dimensions of features, thus improving the quality of features. As shown in Figure 1, Gaussian attention can attend to more important visual clues than the baseline. This is because Gaussian distribution can smooth the distribution of the attention weights, thereby filtering out the respective noisy values. Moreover, since it only involves the mean value and deviations, the speed of forming the attention is lightning-fast. The corresponding extra overhead is negligible. On the other hand, the context information of features is able to enhance the spatial semantics of the ‘Class Token (CLS-Token)’ features. Inspired by ConvNeXt [7], which reparameterizes the transformer architecture into the fully CNN model for efficiency, we design an efficient dynamic convolution module with the context information by using the Gaussian error linear units (GELU) activation function and the layer normalization. This module is also lightly weighted.

In summary, the contributions of this paper are summarized as follows:We propose a novel hybrid model of both transformers and convolution networks, termed TransConv. It improves the robustness of cross-attention with a Gaussian attended MLP module and meanwhile absorbs more semantics via the context-aware dynamic convolution module.TransConv better trades off model performance and efficiency as compared to the state-of-the-arts with a large margin on five datasets.

The rest of this paper is organized as follows: first of all, we review the related work in the Section 2. Then, Section 3 introduces the overall architecture of the proposed TransConv model and each improved module are introduced in detail. Section 4 reports the experimental results on five commonly used datasets and ablation experiments. At last, the conclusion of this paper and the future works are given in Section 5.

## 2. Related Work

In this section, we will introduce the related work in four aspects: unsupervised domain adaptation, vision transformers, dynamic convolution, and contextual information.

### 2.1. Unsupervised Domain Adaptation

From the perspective of using different training methods, there are two main methods of UDA, namely UDA based on metric learning and UDA based on adversarial learning. UDA of metric learning [8,9] mainly measures the distribution difference between different domains by defining a distance metric. UDA can be formulated as a distance minimization problem. For example, the maximum mean discrepancy (MMD) [10] metric has been widely used in UDA methods. UDA of adversarial learning [11,12] mainly trains a domain discriminator and a feature learning network through the adversarial method. The feature learning network learn domain-invariant features and attempt to fool the domain discriminator. When the domain discriminator cannot distinguish whether the input data come from the source domain or the target domain, it will assume that the distributions between the two domains are well aligned. From the perspective of UDA alignment granularity, UDA methods also can be divided into two main methods: domain-level UDA and category-level UDA. Domain-level UDA [13,14] mainly alleviates the distribution difference between the source domain and the target domain by reducing the overall distribution of the source domain and the target domain. Category-level UDA [15,16] mainly achieves more accurate fine-grained alignment by reducing the distribution of each category in the source domain and target domain. The method adopted in this paper is a category-level UDA method based on metric learning. By exploring a hybrid model of transformers and CNNs, our method can fully combine the advantages of both architectures to solve the UDA problem.

### 2.2. Vision Transformer

Transformers [17] were first proposed in the natural language processing (NLP) field and have shown excellent performance in tasks of the NLP field [18,19,20]. As transformers moved from the NLP field to the computer vision field, many studies have shown their effectiveness in computer vision tasks [21,22,23]. ViT [6] was the first work to apply transformers from NLP to computer vision, which is a pure transformer model without convolution. ViT-based variants [24,25] are widely used in image classification and downstream tasks such as object detection [26,27,28], image segmentation [29,30], etc. In the unsupervised domain adaptation task, as compared to the pure convolutional architecture model like ResNet-50, the transformer-type model is better at capturing global features through attention mechanisms. In addition, ResNet-50 relies on the inductive bias for specific images, while transformers have no inductive bias and benefits from large-scale pretraining data. For hybrid-based network models, several studies [31,32] mix transformers with CNN, which further improve the quality of features. This paper also explores the advantage of the hybrid model between ViT and convolutional neural networks from other viewpoints, such as context information and the robustness.

### 2.3. Dynamic Convolution

In traditional regular convolution, the convolution kernel learned is invariant, which leads to performance degradation in the domain shift issue. In contrast, the convolution kernel in dynamic convolution [33,34,35] can be generated dynamically with the input. To better adapt to the problem of domain shift across domains, dynamic convolution is used in our hybrid model instead of regular convolution. Recently, sparse region-based convolutional neural networks (Sparse R-CNN) [28] have also used dynamic convolution to improve the performance of the transformer model architecture in target detection tasks. To better understand the attention mechanism [36], also compared dynamic convolution with regular convolution, deformable convolution, and transformer attention. ConvNeXt explores many strategies from convolutional neural networks to improve performance in the transformer model architecture. In this paper, the proposed dynamic convolution module also borrows some schemes from ConvNeXt to improve the convolutional neural network module.

### 2.4. Contextual Information

Contextual information [37,38] plays a key role in image recognition. Without the help of contextual information, it is easy to identify objects incorrectly. By integrating contextual information, the performance improvement of computer vision systems is very effective. Therefore, compared with the local features of convolutional neural networks, the advantage of the transformer model architecture in improving performance is the use of global context features. The ‘CLS-Token’ features learned in ViT as the global context features are given to the classifier for classification recognition, while the ‘CLS-Token’ features are neglected in Swin transformers. Swin transformers use the global average pooling operation to output global context features for classification recognition. Two ways, in fact, are orthogonal. In this paper, we mix them to form the new dynamic convolution module.

## 3. The Proposed Method

In this section, we first introduce the self-attention module in ViT and the improved Gaussian attended MLP module. After that, we improve the performance of the hybrid model based on ViT by combining contextual information with the dynamic convolution module. Lastly, we introduce our method TransConv, which consists of three parts: a transformer encoder, contextual information combination, and dynamic convolution. The overall structure of the proposed TransConv is shown in Figure 2. The source domain image and the target domain image are respectively split into multiple patches and rearranged by patch embedding to output token features. They are fed into the transformer encoder, where layer normalization serves to normalize, the multi-head attention module adjusts the attention weight of spatial features, and the Gaussian attended MLP module adjusts the attention weight of channel features. The attended features are divided into the ‘CLS-Token’ branch and the average pooling branch, which respectively serve for global class-wise semantics and spatial features. Dynamic convolution makes them adaptive to domain-agnostic. They are concatenated together and then classified by the classifier. Our method simultaneously optimizes the classification (cls) loss and the local maximum mean discrepancy (lmmd) loss. The cls loss and the lmmd loss will be introduced in Section 3.3.

### 3.1. Transformer Encoder

The model designed in MLP-Mixer [39] uses a pure MLP structure with two types of MLP layers, which are channel-mixing MLP and token-mixing MLP. Inspired by MLP-Mixer, TransConv uses the self-attention module in ViT for token mixing and the Gaussian channel mlp module for channel mixing.

**Self-Attention in Transformer.** The basic module of ViT is the self-attention module, referred to as the SA module. The inputs of SA are *Q*, *K*, and *V*, which represent query, key, and value, respectively. To obtain the meaning of each token in the whole image, one dots the product of query with all the transpositions of keys, normalizes the result, and finally uses the softmax function to obtain the weight of the value. In order to provide more possibilities for the self-attention module, multiple self-attention modules are concatenated to form the multi-attention module, referred to as the MSA module.
(1)$$SA\left(Q,K,V\right)=softmax\left(\frac{QK^{T}}{\sqrt{d_{k}}}\right)V$$
where *d* is the dimension of *Q* and *K*.
(2)$$\begin{array}{cc} MSA(Q,K,V) & =Concat\left(head_{1},\cdots ,head_{k}\right)W^{O}\\ where\;head_{i} & =SA(QW_{i}^{Q},KW_{i}^{K},VW_{i}^{V}) \end{array}$$
where $QW_{i}^{Q}$, $KW_{i}^{K}$, $VW_{i}^{V}$ are projections of different heads; $W^{O}$ is a mapping function.

**Gaussian Attended MLP in Transformer** is an improvement of the MLP module in ViT. Gaussian channel attention is an alternative method to improve feature quality, which helps improve performance on UDA tasks and does not require complex training. This is because the Gaussian attended MLP module enhances the denoising ability only using an end-to-end training. In fact, the scaling operation is performed on the MLP module. The attention weights are applied to the channel dimensions in TransConv. This attends important channels and decreases the focus on unimportant ones. Inspired by channel attention methods such as Gaussian context transformer (GCT) [40], the Gaussian attended MLP module is added to the scaling operation of MLP module, and the ‘CLS-Token’ feature is used to calculate weight and adjust the channel dimensions of the input feature, as shown in Figure 3. Specifically, given a feature map *X*∈$R^{B\times HW\times C}$, the global feature can be represented by the learnable ‘CLS-Token’ feature in feature map *X*. ‘CLS-Token’ ∈$R^{B\times 1\times C}$, where *B* is the number of images in a batch, HW is the spatial dimension, and *C* is the channel dimension. First, the ‘CLS-Token’ feature is normalized, which can be expressed as
(3)$$\hat{CLS}=\frac{1}{\sigma }\left(CLS-\mu \right)$$
where μ denotes the mean of the ‘CLS-Token’ feature and σ denotes the variance of the ‘CLS-Token’ feature. Then, the Gaussian function is used to calculate the attention weights:(4)$$G\left(\hat{CLS}\right)=ae^{-\frac{{\left(\hat{CLS}-b\right)}^{2}}{2c^{2}}}$$
where *a* denotes the amplitude of the Gaussian function, *b* denotes the mean of the Gaussian function, and *c* denotes the standard deviation of the Gaussian function.

To simplify the operation, set *a* to constant 1, *b* to constant 0, and *c* to a parameter that can be learned, which can control the channel attention activation. Therefore, the Gaussian function can be simplified to
(5)$$G\left(\hat{CLS}\right)=e^{-\frac{{\left(\hat{CLS}\right)}^{2}}{2c^{2}}}$$

We combine the above operations to form an Gaussian attented MLP, which can be formulated as
(6)$$Y=e^{-\frac{{\left(\frac{1}{\sigma }\left(CLS-\mu \right)\right)}^{2}}{2c^{2}}}X$$
where *X* denotes the input features before the Gaussian attended MLP and *Y* denotes the output features after the Gaussian attended MLP.

**Robustness to Noise.** The pseudo-labeling in the target domain usually contains noises. To further analyze whether the Gaussian channel attention has the ability to denoise the pseudo-labeling, we design an experiment carefully. Specifically, we sample the same number of the same category images from the source and target domains in the W→A task of the Office-31 dataset as the training data, i.e., the training images of the source domain and target domain in each batch belong to the same category. Then, we manually replace the image pairs of the same category with the image pairs of different categories to increase noise and observe the changes in performance and the changes in UDA performance by the ratio of image pairs of different categories, as shown in Figure 4. The *x*-axis represents the ratio of image pairs of different categories in the training data, and the *y*-axis represents the accuracy of different methods on the UDA task. When the *X*-axis is a value of 0.0, it means that all image pairs in a batch have the same category. When the *X*-axis is a value of 1.0, it means that the categories of all image pairs in a batch are different. The red curve represents the results by using the Gaussian attended MLP module, while the blue curve is the results without the Gaussian Attended MLP module. It can be seen that the red curve before the 80 percent ratio of images in different categories achieves better performance than the blue curve, which implies the robustness of the Gaussian attended MLP module to noise.

### 3.2. Dynamic Convolution

Compared with regular convolution, dynamic convolution is more suitable for unsupervised domain adaptation. Because the kernels of dynamic convolution adapt to the input image, dynamic convolution requires an additional convolution kernel generation module. The output features *Y*∈$R^{B\times (HW+1)\times C}$ of the transformer encoder consist of two parts: the features ‘CLS-Token’ ∈$R^{B\times 1\times C}$ and the features *X*∈$R^{B\times HW\times C}$.

**Convolution kernel generation.** First, *X* needs the global average pooling to obtain a global spatial feature, while the ‘CLS-Token’ features is already a global spatial feature, so there is no need to perform additional global average pooling. Then, the global spatial feature is mapped to *K* dimensions through two fully connected (FC) layers, using GELU activation function between two FC layers, and finally, the softmax function is used to complete the normalization. *K* attention weights obtained in this way can be assigned to *K* kernels of this layer. Here, different from the Gaussian attended MLP module, dynamic convolution takes kernels as attention objects.

**Dynamic convolution:** *K* 1 × 1 kernels are convolved with the global spatial feature, and the result of dynamic convolution is obtained by layer normalization (LN), as shown in Figure 5. Finally, to obtain the contextual information, the results of the ‘CLS-Token’ features and features *X* obtained by dynamic convolution are concatenated together and delivered to the classifier for classification. The implementation of dynamic convolution is similar to a dynamic perceptron and can be summarized by the following formula:(7)$$\begin{array}{cc} y & ={\tilde{W}}^{T}\left(x\right)x+\tilde{b}\left(x\right)\\ \tilde{W}\left(x\right) & =\sum_{k=1}^{K}\pi_{k}\left(x\right){\tilde{W}}_{k},\tilde{b}\left(x\right)=\sum_{k=1}^{K}\pi_{k}\left(x\right){\tilde{b}}_{k}\\ 0 & \leq \pi_{k}\left(x\right)\leq 1,\sum_{k=1}^{K}\pi_{k}\left(x\right)=1 \end{array}$$
where $\pi_{k}$ denotes the attention weight of the kth linear function ${\tilde{W}}_{k}^{T}x+{\tilde{b}}_{k}$, which is generated by the convolution kernel generation module and is different for different input *x*.

### 3.3. TransConv: Transformer Meets Convolution

The framework of the proposed TransConv in this paper is shown in Figure 2. It consists of two weight-sharing hybrid models. The hybrid model includes a transformer encoder and dynamic convolution. The source domain images and target domain images in the input are sent to the source domain branch and target domain branch, respectively. In these two branches, the hybrid model participates in learning the representation of a specific domain. In the training phase, the classification result of the source domain is supervised by the labels of the source domain dataset. In the image classification task of UDA, a labeled source domain $D_{s}{\left\{\left(x_{i}^{s},y_{i}^{s}\right)\right\}}_{i=1}^{n_{s}}$ with $n_{s}$ examples and an unlabeled target domain $D_{t}{\left\{x_{j}^{t}\right\}}_{j=1}^{n_{t}}$ with $n_{t}$ examples are provided. The supervised classification loss function $L_{cls}$ for the source domain can be expressed as
(8)$$L_{cls}=\frac{1}{n_{s}}\sum_{i=1}^{n_{s}}J\left(f\left(x_{i}^{s}\right),y_{i}^{s}\right)$$
where *J* (·,·) is the cross-entropy loss function and *f* is the TransConv hybrid network model. However, the target domain has no label, and the classification result of the target domain, namely the pseudo-label, reduces the distribution difference between the source and target domains by minimizing the metric learning loss between the source domain features and target domain features. The domain adaptation metric learning loss selected in this paper is $L_{lmmd}$ [41], which is a loss function for subdomain adaptation. Subdomain adaptation can adjust the subdomain distribution of the source domain and target domain more accurately than global adaptation.
(9)$$\begin{array}{cc} L_{lmmd}=\sum_{l\in L}\frac{1}{C}\sum_{c=1}^{C}[ & \sum_{i=1}^{n_{s}}\sum_{j=1}^{n_{s}}w_{i}^{sc}w_{j}^{sc}k\left(z_{i}^{sl},z_{j}^{sl}\right)\\ \; & +\sum_{i=1}^{n_{t}}\sum_{j=1}^{n_{t}}w_{i}^{tc}w_{j}^{tc}k\left(z_{i}^{tl},z_{j}^{tl}\right)\\ \; & -2\sum_{i=1}^{n_{s}}\sum_{j=1}^{n_{t}}w_{i}^{sc}w_{j}^{tc}k\left(z_{i}^{sl},z_{j}^{tl}\right)] \end{array}$$
where *k* (·,·) is a kernel function [42,43], $z^{l}$ is the lth(l∈L=1,2,⋯,|L|) layer activation, $w_{i}^{sc}$ and $w_{j}^{tc}$ denote the weight of $z_{i}^{sl}$ and $z_{j}^{tl}$ belonging to class *c*, respectively.

To summarize, the objective function of TransConv is
(10)$$L_{cls}+\alpha \cdot L_{lmmd}$$
where α is a hyperparameter. The main steps of our method are reported in Algorithm 1.
**Algorithm 1** TransConv
**Input**: Source and target domain data $X_{s}$ and $X_{t}$; labels for source domain data $y_{s}$.
**Parameter**: parameter α=0.1.**Output**: Predicted labels $y_{t}$ for target domain unlabeled data.
**begin** **while** not converge and epoch < max_epoch **do**
Randomly sample $n_{b}$ labeled source domain instances and $n_{b}$ unlabeled target domain instances.Using the transformer encoder to extract features for images. The multi-head attention module and the Gaussian attended MLP module serve to adjust the attention weight of spatial features Equation  (Equation 2) and of channel features Equation (Equation 6), respectively.Use dynamic convolution to learn two features of the transformer encoder output separately by Equation (Equation 7).Combine the features of the two branches of the dynamic convolution output.Train the classifier and obtain the pseudo-labels of $X_{t}$.Calculate the learning loss in Equation (Equation 10).Update the networks by minibatch SGD.
 **end while**
**end**

## 4. Experiments

To verify the effectiveness of our model, we evaluate our proposed method on four widely used datasets including Office-31, Office-Home, ImageCLEF-DA, and VisDA-2017 for object recognition. MNIST, USPS, and SVHN are used for digit classification. And we compare them with the state-of-the-art UDA methods.

**Digit** classification is an UDA benchmark, consisting of MNIST [44], USPS and Street View House Number (SVHN) [45]. We use the same settings as previous work to train our model, i.e., training phase uses training sets for each pair of source and target domains, and the testing phase uses the test set for target domain to perform evaluations.

The **Office-31** dataset [46] contains 4652 images in 31 categories and consists of three domains: Amazon (A), DSLR (D), and Webcam (W). Amazon (A) contains 2817 images, which were downloaded from www.amazon.com; 498 images in DSLR (D) and 795 images in Webcam (W) were captured in an office environment by web and digital SLR cameras, respectively.

The **Office-Home** dataset [47] consists of 15,588 images in 65 object categories. It contains images from four different domains: artistic images (A), clip art (C), product images (P), and real-world images (R). Images in each domain are collected in office and home environments. There are 2427 images in (A), 4365 images in (C), 4439 images in (P), and 4357 images in (R), respectively.

The **ImageCLEF-DA** dataset contains 1800 images in 12 categories. It consists of three domains: Caltech-256 (C), ImageNet ILSVRC 2012 (I), and Pascal VOC 2012 (P). There are 600 images in each domains and 50 images in each category.

The **VisDA-2017** dataset [48] contains about 280k images in 12 categories. It includes three domains: training, validation, and test domains. It is a dataset from simulation to a real environment. The training set has 152,397 images, which were generated by the same object under different circumstances, the 55,388 images in the validation set and the 72,372 images in the test set are real-world images.

**Baseline Methods** For Digital dataset, we compare TransConv with DANN [49], ADDA [12], SHOT [50], DSAN [41], CDAN [11], MCD [51] and TVT [5]. For Office-31 dataset, we compare TransConv with ResNet-50 [1], DANN, CDAN+E [11], SHOT, ALDA [52], DSAN, ALSDA [53], PICSCS [54], TVT and CDTrans [4]. For Office-Home dataset, we compare TransConv with ResNet-50, SHOT, ALDA, CDAN+E, DSAN, ALSDA, PICSCS, TVT and CDTrans. For ImageCLEF-DA dataset, we compare TransConv with ResNet-50, DANN, CDAN+E, DSAN, PICSCS, DALN [55] and MCC+NWD [55]. For VisDA-2017 dataset, we compare TransConv with ResNet-50, DANN, CDAN+E, SHOT, DSAN, ALDA, ALSDA, TVT, and CDTrans. The results of most baselines are extracted from [5,41]. For the rest, we refer to the results in their original articles.

**Implementation Details** The ViT-B/16 model pretrained on ImageNet 21k is used as a backbone network to extract image features. The input image size in our experiments is 256 × 256, and the size of each patch is 16 × 16. The transformer encoder of ViT-B/16 consists of 12 transformer encoder layers. We train the model using a minibatch stochastic gradient descent (SGD) optimizer with a momentum of 0.9, and we initialize the learning rate as 0. We linearly increase its learning rate to 3 $\times 10^{-2}$ after 500 training steps, and then decrease it by the cosine decay strategy. Experiments are conducted on a single card 2080 Ti with 11 G memory. The batch size is set to 16.

**Results of Digit Recognition** The classification results of the three tasks in digital recognition are shown in Table 1. Since current compared methods only evaluate three cases (i.e., SVHN→MNIST, USPS→MNIST and MNIST→USPS), and there is no comparisons for the remaining three cases, we also use the same settings as the previous studies. TransConv achieves the same best accuracy as TVT on MNIST→USPS task, and 0.2% lower than the best average classification accuracy. The above-mentioned results demonstrate the effectiveness of the TransConv model and well alleviate the domain shift problem.

**Results of Object Recognition.** We evaluate four datasets for object recognition tasks, including Office-31, ImageCLEF-DA, Office-Home, and VisDA-2017. The results of object recognition are shown in Table 2, Table 3, Table 4 and Table 5. In Table 3, TransConv achieves the best average classification accuracy on ImageCLEF-DA, and achieves the significant improvement over the best prior UDA method (92.3% vs. 91.3%). But TransConv is lower than the best prior UDA method on Office-31, Office-Home and VisDA-2017. In Table 2 and Table 4, TransConv is lower than TVT on Office-31 (92.8% vs. 93.9%) and Office-Home (82.9% vs. 83.6%). In Table 5, TransConv is lower than CDTrans on VisDA-2017 (80.9% vs. 88.4%). In Table 2, it can be seen from the difference in the number of samples and the results obtained in the three domains (Amazon, DSLR, and Webcan) of the Office-31 dataset that the larger the source domain dataset, the higher the corresponding performance. Moreover, as shown in Table 6, TransConv surpasses the Baseline (92.8% vs. 91.7%). This is also evidenced by the *t*-SNE visualization of learned features as shown in Figure 6. We visualize the network activations of baseline and TransConv for task A→W of Office-31 dataset. Red points are source samples and blue are target samples. Figure 6a shows the result for baseline, we can find that the source and target domains are not aligned very well and some points are hard to classify. In contrast, Figure 6b shows the our TransConv. It is observed that the source and target domains are aligned very well. The experimental results show that the hybrid model using the Gaussian attended MLP module with denoising capability and highly efficient dynamic convolution module can improve the domain adaptation problem to some extent. TransConv is an effective attempt to the hybrid model of transformers and convolutional neural networks.

**Ablation Study.** In order to learn the individual contribution of Gaussian attended MLP, dynamic convolution and context information in improving the knowledge transferability of ViT, we conduct the ablation study, as shown in Table 6. Compared to the TransConv model, the Gaussian attended MLP, dynamic convolution, and context information are subtracted, respectively. Without the Gaussian attended MLP, the average classification accuracy is reduced by 0.2%; without dynamic convolution, the average classification accuracy is reduced by 0.5%; and without the context information, the average classification accuracy is reduced by 0.6%. Baseline is the result of simultaneously being without the Gaussian attended MLP, dynamic convolution, and contextual information, which reduces the average classification accuracy by 1.1%, indicating the significance of the three improvement methods in the model performance. To understand the effect of each improvement in dynamic convolution, we conduct the ablation study, as shown in Table 7; compared to the TransConv model, without LN, the average classification accuracy is reduced by 0.4%, and without GELU, the average classification accuracy is reduced by 0.1%. Baseline is the result of without LN and without GELU at the same time, which reduces the average classification accuracy by 0.2%, indicating that the two improvements play a role in the model performance.

**Parameter Sensitivity and Robustness.** In our model, the hyperparameter α controls the $L_{lmmd}$. To better understand the effects of α, we report the sensitivities of α in Figure 7a. It can be seen that our TransConv achieves the best results when α = 0.1. Therefore, this article fixes α = 0.1. We also observe the robustness of our model by changing the distribution of the source domain and target domain. In Figure 7b, the *d* represents the interdomain distance, ‘-’ represents reducing the interdomain distance, and ‘+’ represents increasing the interdomain distance. It can be seen that when the increasing/decreasing distance is less than *d*, the model performance decreases. When increasing the distance greater than *d*, the larger the distance, the more the model performance decreases.

## 5. Discussion

To see the efficiency of TransConv, we especially compare it with TVT and CDTrans, as shown in Table 8. First, we compare TransConv with TVT. TVT uses a pure transformer architecture with adversarial training, while TransConv uses a hybrid architecture that is easy to implement without adversarial training. This also saves some overheads. In addition, TVT has multiple loss functions, while TransConv only uses two loss functions. Moreover, TransConv has only one hyperparameter, while TVT has three hyperparameters. TransConv has fewer loss functions and hyperparameters, thereby avoiding hyperparameter tuning. Second, we compare TransConv with CDTrans. CDTrans uses a three-branch pure transformer architecture, which requires a large amount of cross-attention computations and two different training phases. Our hybrid architecture does not require a large number of cross-attention calculations and only needs to be trained in an end-to-end single-phase way. TransConv balances model improvement and computation overheads. Overall, these results verify the advantages of TransConv.

## 6. Conclusions

In this paper, we tackle the problem of unsupervised domain adaptation by improving transformer encoders and using context information in a novel hybrid way. This induces a new hybrid network structure, TransConv. Specifically, TransConv can improve the robustness of the features through a Gaussian attended MLP module and can improve semantics of local features by context-aware dynamic convolution. Experimental results on widely used benchmarks demonstrate the effectiveness of the TransConv model. Future work will further investigate other strategies to efficiently achieve the state-of-the-arts (SOTA) performance on UDA tasks. TransConv is restricted to inadequate transferability of global class-wise semantics and global spatial representations because feature confusion needs more fine-grained feature interaction.

## Figures and Tables

**Figure 1 entropy-26-00469-f001:**
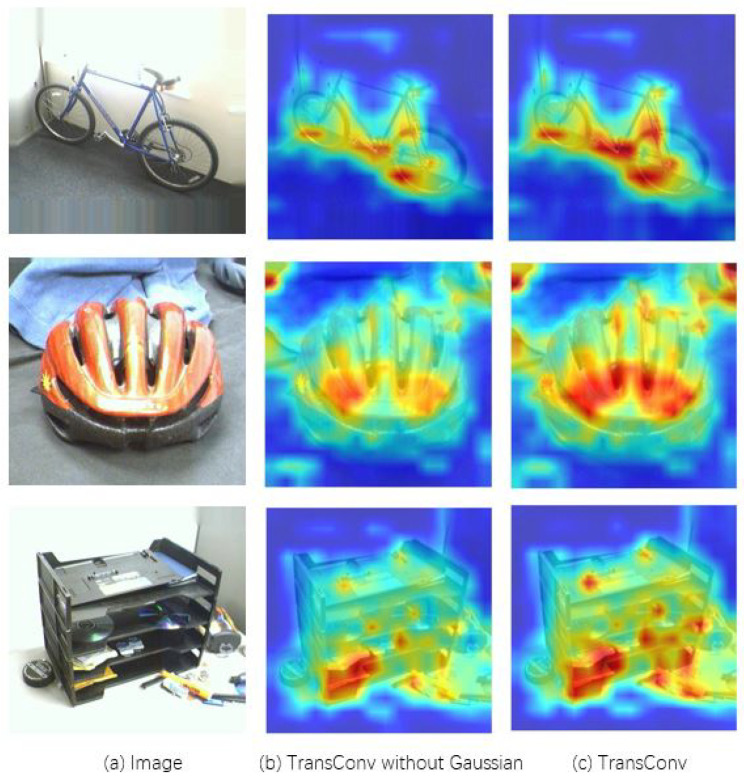
Attention visualization of bike, bike helmet, and letter tray in the Office-31 dataset. The hotter the color, the higher the attention.

**Figure 2 entropy-26-00469-f002:**
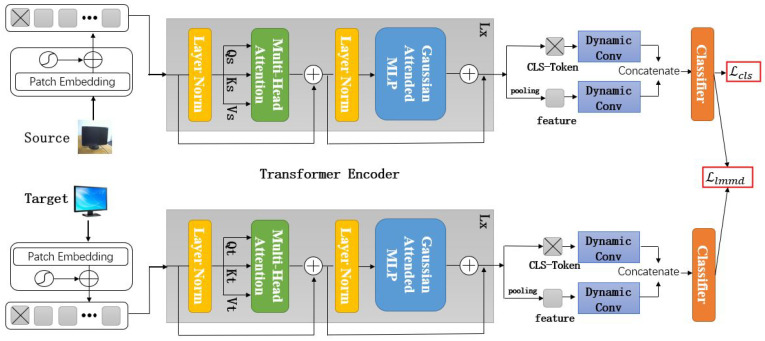
The proposed TransConv framework.

**Figure 3 entropy-26-00469-f003:**
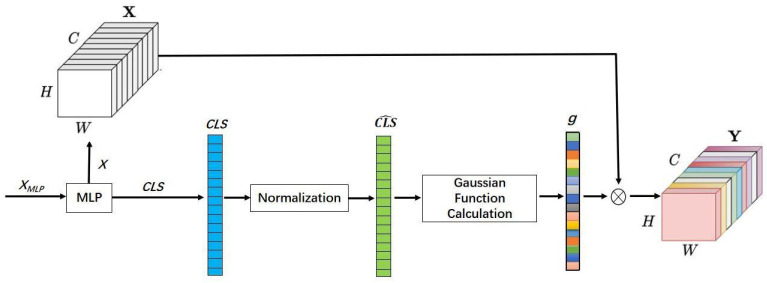
The Gaussian attended MLP framework. The MLP is a scaling operation implemented by two FC modules. For the input $X_{MLP}$ of MLP, its dimension is $R^{B\times (HW+1)\times C}$. For the output of MLP, the dimension of *X* is $R^{B\times HW\times C}$ and the dimension of CLS is $R^{B\times 1\times C}$. The normalization module normalizes the CLS feature to obtain $\hat{CLS}$ and the Gaussian function calculation module calculates the attention weight for $\hat{CLS}$ to obtain *g*. The *g* represents the attention activations. ⨂ denotes broadcast element-wise product.

**Figure 4 entropy-26-00469-f004:**
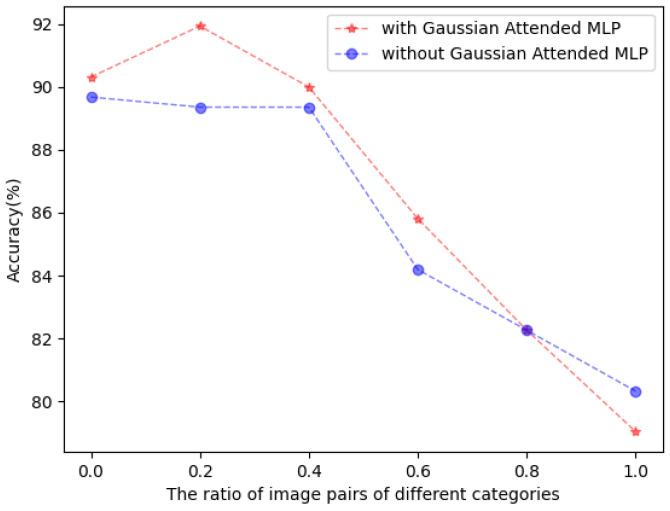
The model with Gaussian Attended MLP modules vs. without Gaussian Attended MLP modules. The red/blue curves represent the model with and without the Gaussian Attended MLP modules.

**Figure 5 entropy-26-00469-f005:**
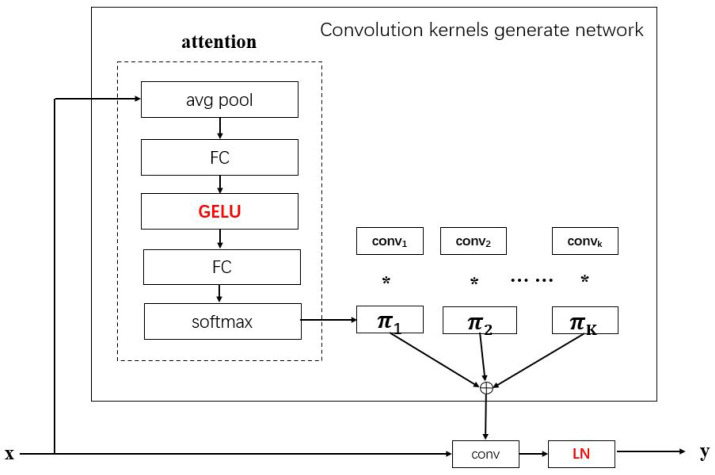
Improved dynamic convolution module framework. The red represents the improved part based on the original dynamic convolution.

**Figure 6 entropy-26-00469-f006:**
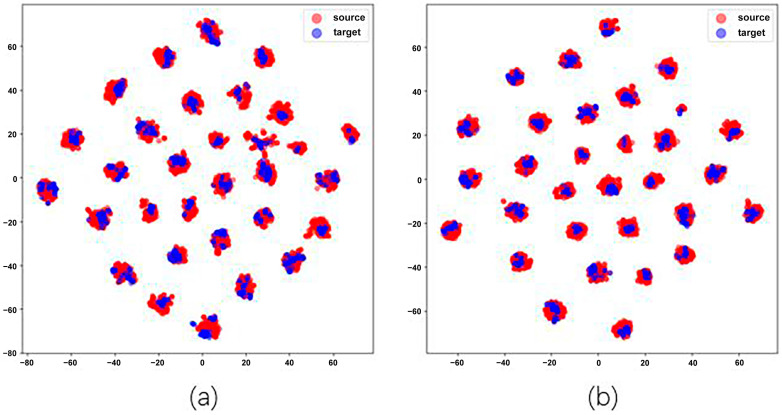
Feature visualization of (**a**) the baseline and (**b**) TransConv using *t*-SNE on the task A→D of Office-31 dataset, where red and blue points indicate the source and the target domain, respectively.

**Figure 7 entropy-26-00469-f007:**
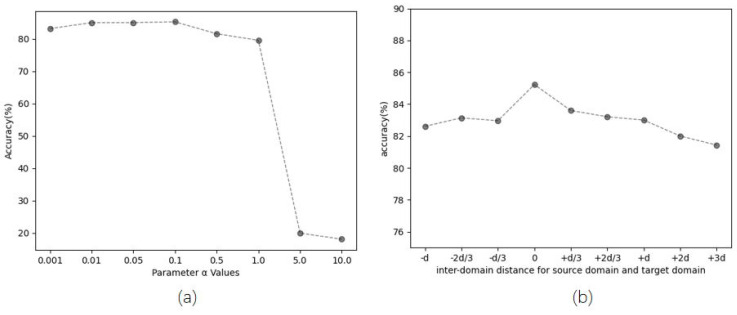
Model analysis on evaluation W→A. (**a**) Parameter sensitivity of α. (**b**) The changes of performance by the interdomain distance for source domain and target domain.

**Table 1 entropy-26-00469-t001:** Performance comparison on digits dataset. The best performance is marked as bold.

Method	S→M	U→M	M→U	Avg
DANN	73.9	73.0	77.1	74.7
ADDA	76.0	90.1	89.4	85.2
SHOT-IM	89.6	96.8	91.9	92.8
DSAN	90.1	95.3	96.9	94.1
CDAN	89.2	98.0	95.6	94.3
MCD	96.2	94.1	94.2	94.8
TVT	**99.0 **	**99.4**	**98.2**	**98.9**
TransConv	98.7	99.2	**98.2**	98.7

**Table 2 entropy-26-00469-t002:** Performance comparison on Office-31 dataset.

Method	A→W	D→W	W→D	A→D	D→A	W→A	Avg
ResNet-50	68.4	96.7	99.3	68.9	62.5	60.7	76.1
DANN	82.0	96.9	99.1	79.7	68.2	67.4	82.2
CDAN+E	94.1	98.6	**100.0**	92.9	71.0	69.3	87.7
SHOT	90.1	98.4	99.9	94.0	74.7	74.3	88.6
ALDA	95.6	97.7	**100.0**	94.0	72.2	72.5	88.7
DSAN	93.6	98.3	**100.0**	90.2	73.5	74.8	88.4
PICSCS	93.2	99.1	**100.0**	93.6	77.1	78.0	90.2
ALSDA	95.2	99.2	**100.0**	95.8	78.1	77.5	91.0
TVT	96.4	**99.4**	**100.0**	96.4	**84.9**	**86.1**	**93.9**
CDTrans	**96.7**	99.0	**100.0**	**97.0**	81.1	81.9	92.6
TransConv	94.8	99.1	99.8	93.2	84.5	85.2	92.8

**Table 3 entropy-26-00469-t003:** Performance comparison on ImageCLEF-DA dataset.

Method	I→P	P→I	I→C	C→I	C→P	P→C	Avg
ResNet-50	74.8	83.9	91.5	78.0	65.5	91.2	80.7
DANN	75.0	86.0	96.2	87.0	74.3	91.5	85.0
CDAN+E	77.7	90.7	97.7	91.3	74.2	94.3	87.7
DSAN	80.2	93.3	97.2	93.8	80.8	95.9	90.2
DALN	80.5	93.8	97.5	92.8	78.3	95.0	89.7
MCC+NWD	79.8	94.5	**98.0**	94.2	80.0	97.5	90.7
PICSCS	81.9	94.8	96.8	95.8	**81.7**	96.5	91.3
TransConv	**83.2**	**96.7**	97.7	**97.7**	80.5	**97.7**	**92.3**

**Table 4 entropy-26-00469-t004:** Performance comparison on Office-Home dataset. * indicates the results of using the ensemble learning strategy.

Method	Ar→Cl	Ar→Pr	Ar→Rw	Cl→Ar	Cl→Pr	Cl→Rw	Pr→Ar	Pr→Cl	Pr→Rw	Rw→Ar	Rw→Cl	Rw→Pr	Avg
ResNet-50	44.9	66.3	74.3	51.8	61.9	63.6	52.4	39.1	71.2	63.8	45.9	77.2	59.4
SHOT	57.1	78.1	81.5	68.0	78.2	78.1	67.4	54.9	82.2	73.3	58.8	84.3	71.8
ALDA	53.7	70.1	76.4	60.2	72.6	71.5	56.8	51.9	77.1	70.2	56.3	82.1	66.6
CDAN+E	54.6	74.1	78.1	63.0	72.2	74.1	61.6	52.3	79.1	72.3	57.3	82.8	68.5
DSAN	54.4	70.8	75.4	60.4	67.8	68.0	62.6	55.9	78.5	73.8	60.6	83.1	67.6
PICSCS	56.0	79.0	81.0	67.6	81.3	79.9	68.4	55.0	82.4	72.3	58.5	85.0	72.2
ALSDA	60.4	78.9	81.9	67.8	77.4	77.5	68.6	57.9	83.0	80.1	62.8	85.0	73.4
TVT	**74.9**	86.8	**89.5**	**82.8**	**88.0**	88.3	79.8	**71.9**	**90.1**	**85.5**	**74.6**	90.6	**83.6**
CDTrans	68.8	85.0	86.9	81.5	87.1	87.3	79.6	63.3	88.2	82.0	66.0	90.6	80.5
TransConv	68.2	86.7	88.4	82.4	87.4	88.2	81.8	69.7	89.6	84.0	73.1	**91.1**	82.6
TransConv *	69.9	**87.1**	88.6	82.6	87.5	**88.4**	**82.1**	70.2	89.8	84.6	73.1	**91.1**	82.9

**Table 5 entropy-26-00469-t005:** Performance comparison on VisDA-2017 dataset.

Method	Plane	Bcycl	Bus	Car	Horse	Knife	Mcycl	Person	Plant	Sktbrd	Train	Truck	Avg
ResNet-50	55.1	53.3	61.9	59.1	80.6	17.9	79.7	31.2	81.0	26.5	73.5	8.5	52.4
DANN	81.9	77.7	82.8	44.3	81.2	29.5	65.1	28.6	51.9	54.6	82.8	7.8	57.4
CDAN+E	85.2	66.9	83.0	50.8	84.2	74.9	88.1	74.5	83.4	76.0	81.9	38.0	73.9
SHOT	94.3	88.5	80.1	57.3	93.1	93.1	80.7	80.3	91.5	89.1	86.3	58.2	82.9
DSAN	90.9	66.9	75.7	62.4	88.9	77.0	**93.7**	75.1	92.8	67.6	89.1	39.4	75.1
ALDA	93.8	74.1	82.4	69.4	90.6	87.2	89.0	67.6	93.4	76.1	87.7	22.2	77.8
ALSDA	93.8	72.8	81.0	49.0	82.9	90.5	89.3	80.8	88.5	86.6	87.3	43.9	78.9
TVT	92.9	85.6	77.5	60.5	93.6	98.2	89.4	76.4	93.6	92.0	91.7	55.7	83.9
CDTrans	97.1	**90.5**	82.4	77.5	96.6	96.1	93.6	**88.6**	**97.9**	86.9	90.3	**62.8**	**88.4**
TransConv	**97.8**	88.4	85.1	**78.1**	96.7	**98.9**	92.1	0.0	96.1	96.6	**96.6**	38.8	80.4
TransConv *	97.6	87.2	**88.9**	72.6	**96.8**	98.7	93.3	0.1	96.1	**96.9**	96.3	46.4	80.9

**Table 6 entropy-26-00469-t006:** Ablation study of TransConv on Office-31 dataset.

Method	A→W	D→W	W→D	A→D	D→A	W→A	Avg
ViT(baseline)	93.5	**99.2**	99.3	91.8	84.1	82.2	91.7
TransConv without Gaussian	94.1	**99.2**	**100.0**	**93.2**	**84.6**	84.4	92.6
TransConv without Dynamic	94.1	99.0	99.8	93.0	84.0	83.7	92.3
TransConv without Context	**95.2**	99.1	99.8	92.6	83.5	83.0	92.2
TransConv	94.8	99.1	99.8	**93.2**	84.5	**85.2**	**92.8**

**Table 7 entropy-26-00469-t007:** Ablation study of dynamic convolution on Office-31 dataset.

Method	A→W	D→W	W→D	A→D	D→A	W→A	Avg
Dynamic (baseline)	**95.2**	**99.2**	**99.8**	92.4	84.4	84.5	92.6
TransConv without LN	94.7	**99.2**	**99.8**	92.0	84.1	84.7	92.4
TransConv without GELU	94.8	99.1	**99.8**	**93.6**	84.2	84.8	92.7
TransConv	94.8	99.1	**99.8**	93.2	**84.5**	**85.2**	**92.8**

**Table 8 entropy-26-00469-t008:** Performance comparison of TransConv, TVT, and CDTrans on the Office-31 dataset. The running time is the convergence time and is measured on a single 2080Ti GPU.

Method	Hyper-Params	Loss	Adversarial	FLOPs	Running Time	Params	Accuracy
TVT	3	4	Yes	22.0 G	2442 s	86.5 M	93.9
CDTrans	1	2	No	33.0 G	714 s	173.2 M	92.6
TransConv	1	2	No	22.0 G	1372 s	88.8 M	92.8

## Data Availability

Data are contained within the article.

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
