# Peer review of "TransConv: Transformer Meets Contextual Convolution for Unsupervised Domain Adaptation"

_entropy, 2024, doi:10.3390/e26060469_

Round 1
Reviewer 1 Report
Comments and Suggestions for Authors
The proposed framework towards unsupervised domain adaptation is sound, including a few architectural contributions, consisting of Gaussian Attended MLP in combination to dynamic convolutions. Still, I would like to ask the authors to extend the discussion section by providing more intuitions regarding their model performance in comparison to TVT and CDTrans. Which architectural decisions are pivotal for certain classes? Also, they could support their observations using explainability techniques for increasing the readers' awareness over the improvement points. Finally, please add a conclusions section, indicating also the limitations of your approach.
Comments on the Quality of English LanguageThe use of English language is fine, but there are just a few minor mistakes, for example in the sentence in section 3.3, "In the image classification task of UDA..." the verb is missing, so please do a thorough grammatical check of your paper before the final submission.
Reviewer 2 Report
Comments and Suggestions for Authors
(1) This paper (# entropy-2993743) hybridizes two approaches for UDA to reduce the cost of conventional disadvantages such as high computational overhead or complex training processes. Therefore, what is expected in the paper is how much this hybrid strategy has contributed to reducing the aforementioned cost. A reinforcement from this perspective seems necessary.
(2) In addition, to improve the originality of the proposed method, it would also be good to compare 'robustness' while artificially varying the distribution or size of source and target datasets in addition to the comparison of classification accuracy alone.
(3) The method is well summarized by Algorithm 1. In order to more clearly explain the operational excellence of the algorithm, the following may be considered.
(i) It seems that the specific tasks performed in each step should be accurately defined through pseudo-codes. In particular, what should be processed in Step 2 may be expressed more clearly.
(ii) In general, a new algorithm should be accompanied by an analysis of time complexity as well as output performance. This analysis may also be used for comparison of computational overheads.
(iii) Also of interest is how sensitive meta-parameters (if present, e.g., alpha here) are to the performance of the algorithm. Therefore, the consideration of the above items will be of great help to practitioners interested in using the algorithm.
(iv) Algorithm 1 should be referred to in the text.
(4) In order to increase the readability of this paper, it seems necessary to revise it in several points, including:
(i) The role of the symbols and boxes shown in Figure 2 needs to be explained in the text or caption. In the same context, it would be nice if it was indicated that the cls loss and the lmmd loss will be defined later (in fact, they are defined in the following Section 3.3).
(ii) Figure 3 also needs to be improved by explaining the functions of its components and their roles in the caption. For example, it is necessary to indicate what the input/output data of MLP is and what the dimensions are.
(iii) In Figure 4, it would be nice to add a description of what happens when the x-axis is a value of 0.0 or 1.0.
(iv) In Table 1, one sentence about why only three (i.e., SVHN→MNIST, USPS→MNIST, and MNIST→USPS) of the total six cases were evaluated may help readers.
(v) In Tables 2, 6, and 7, the number of samples of the three domains (Amazon, DSLR, and Webcan) of the Office-31 dataset varies greatly, 2817, 498, and 795, respectively. It would be nice if the effect of the size of the source dataset on the experimental results was considered.
(vi) In Table 4, it is shown that TVT [5] achieves 83.6% accuracy in Avg, while TransConv* achieves 82.9%. Comments on these results will help readers understand the superiority of the proposed method.
(vii) The abbreviations used in the text are kindly summarized in a table. Nevertheless, other abbreviations are used without definition or full spelling, which also degrades the readability of the paper.
(viii) Typos: attetion -> attention (Line 213); "In Proceedings of the Proceedings of the ..." that has been repeated several times (Lines 344, 348, ... ).
(5) Finally, if any limitations of the proposed algorithm have been discovered at this point, a detailed mention of them would be of great help to researchers interested in this methodology.
Round 2
Reviewer 2 Report
Comments and Suggestions for Authors
In the first round, several opinions were presented to clarify the excellence of the hybrid strategy and to increase the readability of the paper. In response to these opinions, many revisions were made: a new chapter was included, and supplements were also made in tables and pictures. Thank you for the hard work of the authors. Although it is a very minor problem, however, it would be nice if it was revised before going further. In Fig. 3, it seems that the MLP's input should be additionally marked (refer to the answer to the comment (4-ii)). A typo: "... the X-axis value is is a value ...".
